# Compositional Characteristics, Fatty Acid Profile, Phenolic Content and Volatile Organic Compounds (VOCs) of Feta Cheese Made in Mountainous Grasslands and Plains of Greece

Martha Maggira [1], Maria D. Ioannidou [1], Zoi M. Parissi [2], Eleni M. Abraham [2], Maria Karatassiou [3] and Georgios Samouris [1,*]

1   Department of Hygiene and Technology of Food of Animal Origin, Veterinary Research Institute, Hellenic Agricultural Organization-DIMITRA, Campus of Thermi, 57001 Thessaloniki, Greece; ioannidou@vri.gr (M.D.I.)
2   Laboratory of Range Science (236), School of Forestry and the Natural Environment, AUTh, 54124 Thessaloniki, Greece; pz@for.auth.gr (Z.M.P.); eabraham@for.auth.gr (E.M.A.)
3   Laboratory of Range Ecology (286), School of Forestry and the Natural Environment, AUTh, 54124 Thessaloniki, Greece; karatass@for.auth.gr
*   Correspondence: samouris@vri.gr

**Abstract:** The study aimed to compare the physicochemical composition, fatty acid profile, phenolic content and volatile organic compounds (VOCs) of feta cheese produced in two regions in Greece (mountainous grasslands and plains). The VOC profiles of forage in the mountainous grasslands were also investigated in order to detect any relationship between the two profiles. Forbs were found to be the major functional group in the mountainous area and the most abundant VOCs were aldehydes, alcohols, and terpenes. Significant variations were observed between the main compositional features of the two types of cheeses, with the cheese produced from the milk of ewes and goats grazing in the mountainous region having a higher moisture content, lower fat content, and higher protein content. A total of 34 VOCs were identified and the most abundant groups in the cheese samples were esters, hydrocarbons, and alcohols. The fatty acid profile, phenolic content, and terpene profile of the cheese were also affected by the type of pasture. The cheese produced from the milk of ewes and goats grazing in the mountainous region also had a higher content of conjugated linoleic acid (CLA) and a lower content of saturated fatty acids (SFAs). These findings suggest that the grasslands grazed by ewes and goats in the mountainous region have a positive impact on the nutritional quality of feta cheese.

**Keywords:** fatty acids; feta cheese; physicochemical characteristics





## 1. Introduction

One of the most widely consumed products manufactured in the Greek dairy sector is feta cheese. Feta cheese is revered as a typical Greek brined white cheese with a long history that dates back to ancient times. It is known for its salty, slightly acidic flavor and highly regarded organoleptic qualities [1]. The composition and sensory properties of feta cheese can vary significantly, as it is made from either 100% sheep milk or a blend of sheep and up to 30% goat milk [2]. Many factors, such as animal breed, feeding practices, season and milking time, as well as climatic conditions, and rangelands floristic composition, affect the composition and sensory attributes of the milk. Consequently, the quality of the produced cheese [3] has been affected.

Due to seasonality and considerable climatic variability, the vegetation in Mediterranean grassland ecosystems is highlighted by a high species diversity and richness [4]. Transhumance has taken use of the high floristic diversity of grasslands because it is a long-standing method of living with spatial and temporal environmental ambiguity [5,6]. Thus, the dietary requirements of the grazed animals could have a significant effect on

the quality of dairy products. Moreover, the botanical composition of the grasslands that animals are grazing on has a significant impact on the terpene content and fatty acid profile of bovine milk and, therefore, in cheese [7–9]. The fatty acid composition of the milk used in dairy product production directly affects their quality attributes and subsequent consumer acceptance [10]. Animal diet could influence the bacterial composition of milk as well as the profile of milk constituents, including fat, proteins, flavor compounds, and other minor components [11]. By understanding and managing animal feeding, it becomes possible to influence the overall quality of cheese and its sensory characteristics.

The characteristics of plant species are frequently reflected in the sensory qualities of cheeses made in the mountains during the spring and summer [11]. It has been discovered that milk from sheep and goats that grazed has a better composition of beneficial bioactive fatty acids (such as CLA cis-9, trans-11, and n-3 fatty acids) and volatile aromatic compounds like terpenes [12].

The present research involved producing feta cheese during the summer period using milk from transhumant flocks of ewes and goats that exclusively graze on rangelands. This study aims to investigate the physicochemical composition, fatty acid profile, phenolic content, and terpenes of traditional feta cheese produced by small ruminants that graze in a mountainous and a lowland region of Greece. By comparing the cheeses from both locations, the study attempts to determine compositional similarities and differences among them. In addition, obtaining the terpene profile of the forage in the mountainous grassland will provide insights into the botanical composition of the grazing lands.

## 2. Materials and Methods

### 2.1. Chemicals and Reagents

The reagents employed in the analyses included sulfuric acid ($H_2SO_4$) 98% for analysis, and boric acid ($H_3BO_3$) (from PanReac AppliChem, Darmstadt, Germany), ammonium iron (III) sulfate 12 hydrate and potassium hydroxide (KOH) 85% (from Panreac, Barcelona, Spain), sodium hydroxide (NaOH) (from Merck, Darmstadt, Germany), silver nitrate ($AgNO_3$) 0.1 mol (from Fixanal, Fluka, Munchen, Germany), sodium bisulfate ($NaHSO_4$) anhydrous (from Fluka Analytical, Munchen, Germany), sodium sulfate anhydrous ($Na_2SO_4$) (from Lach-Ner, Neratovich, Czech Republic), hydrochloric acid (HCl) 0.1 N (from VWR Chemicals BDH, Rosny-sous-Bois cedex, France), methanol (VWR chemicals, Radnor, PA, USA) and amylalcohol ($C_5H_{11}OH$), ammonium thiocyanate ($NH_4SCN$) 0.1 N, sodium hydroxide 0.1 mol/L, nitric acid ($HNO_3$) 65%, and n-hexane pesticide grade (from ChemLab, Zedelgem, Belgium). Tashiro's indicator solution was purchased by Honeywell Fluka (Munchen, Germany) and the Cu tablets used for Kjeldahl from Gerhardt (Königswinter, Germany). For lipid oxidation, butylated hydroxytoluene was purchased from Sigma Aldrich (Darmstadt, Germany), as well as 2-thiobarbituric acid (Sigma-Aldrich, Darmstadt, Germany), and trichloroacetic acid (Merck, Darmstadt, Germany).

The reference standards used in gas chromatography included a 37-component mixture of fatty acids methyl esters (FAME), a 14-polyunsaturated fatty acids mixture No. 2 (PUFA No. 2), a 13-polyunsaturated fatty acids mixture No. 1 (PUFA No. 1), and a Linoleic acid, conjugated methyl ester standard (CLA) purchased from Sigma Aldrich (St. Louis, MO, USA).

### 2.2. Animals and Diet

Three upland meadows in the villages of Polyneri (G1) and Aetia (G2, G3), which are part of the Grevena Regional Unit in northwest Greece, were the study's locations. According to the Köppen–Geiger system, the climate is classified as Dfb, and the research sites are elevated to a height of around 1100 m above sea level [13]. The average yearly temperature was 9.9 °C, with 779 mm of precipitation. These grasslands were grazed by transhumant flocks of ruminants from April to October, primarily sheep in Aetia and cattle and sheep in Polyneri. In the grasslands, vegetation cover was measured with the line and point method [13]. Three experimental transects (25 m each) were established in each

grassland, and floristic composition was calculated and classified into five functional plant groups: (1) grasses, (2) legumes, (3) forbs, (4) woody and (5) graminoids. In each grassland, forage production samples were collected at the inflorescence stage during June, using two sampling quadrats of 0.5 × 0.5 m in each transect at 8 and 16 m (i.e., six quadrats per grassland). After that, the collected forage production was separated by hand into the aforementioned five functional plant groups.

During their stay in the plains for the time period of November to March, flocks and herds grazed in artificial pastures in close proximity to their farms and were also offered commercial mixtures of concentrates and vitamin and mineral premixes on a farm. Concentrates' mixtures and premixes were selected according to the productive stage of the animals, in order to sufficiently cover their nutritional needs.

### 2.3. Cheesemaking

Milk from flocks that grazed in mountainous grasslands in Western Macedonia (Area M), Greece, was heated to 50 °C for bactofugation (no starter culture was added), cooled to 36 °C, and then coagulated by adding commercial animal/calf rennet (CHR-Hansen, Hørsholm, Denmark) (coagulation took place in 40–45 min).

Curd was divided into 1–3 cm cubes, allowed to rest for 15 min, placed in drainage molds with perforated inner walls and bottoms, and then pressed by a weight equal to the weight of the curd. After the dry surface was salted with coarse salt so that the final salt content in the cheese mass was around 3–3.5% NaCl, the cheese blocks were kept at 16 °C for 5 days. More specifically, in the first salting, it remained for 2 h and was first turned upside-down; in the second salting, it remained for 2 h, was turned upside-down for the second time, remained for 20 h, and removed from turn over molds; and in the third salting, it was placed in temporary packaging in containers filled with brine 14–16 g/100 g NaCl where it remained for 2 days and then left for 10 more days at 15–18 °C. The cheese blocks were then finally packaged and placed in containers with brine solution (6.7% NaCl) to continue maturing for the remaining two months in cold storage at 2–5 °C. Feta cheese can be consumed after ripening for at least two months, as to Codex of Foodstuff and Drinks requirements.

The same procedure was conducted with milk from flocks that grazed in plains of Western Macedonia, Greece (Area P).

### 2.4. Physicochemical Analysis

Cheese samples were examined for moisture, pH, protein, fat content and NaCl content. After drying cheese samples (2–5 g) at 105 °C until reaching a constant weight, the moisture content was assessed [14]. Results were expressed as a percentage (%). The Gerber-Van Gulik technique was used to calculate the fat content of cheese samples [15]. The amount of fat in dry matter (FDM) was calculated from Equation (1), using the obtained values for Fat (%) and Moisture (%).

$$\text{FDM} = \frac{\%\text{Fat}}{(100 - \%\text{Moisture})} \tag{1}$$

Following the Kjeldahl method's assessment of total nitrogen [16], the protein content of cheese samples was determined (TN*6.38). Cheese nitrogen fractions were obtained according to IDF [17]. The NaCl content was estimated using the modified Volhard method [18], whereas the salt-in-moisture (S/M) content was also determined. All of the parameters that were taken into account underwent duplicate analysis.

### 2.5. Lipid Oxidation

Malondialdehyde (MDA) generation was measured using a selective third-order derivative spectrophotometric approach to evaluate lipid oxidation [19]. Based on the height of the third-order derivative peak at 521.5 nm and the slope and intercept data of the computed least squares fit of a standard calibration curve made using 1,1,3,3-

tetraethoxypropane, the concentration of MDA in the samples under analysis was determined. Nanograms of MDA per gram of cheese is the unit of measurement for lipid oxidation.

### 2.6. Fatty Acids Analysis

According to the ISO 12966-2:2017 method (with slight modifications applied by Zeppa et al. [20]), fatty acid methyl esters were prepared through transesterification with potassium hydroxide. Milk fat underwent methylation in accordance with the earlier protocol [21].

An HP 6890 gas chromatograph fitted with a flame ionization detector (FID) and a split/splitless injector (in split mode) was used to analyze the methyl esters of fatty acids. Samples were injected using 1 μL split injections with a split ratio of 50:1 onto a DB-23 (60 m × 0.25 mm × 0.25 μm) column (J & W Scientific, Inc., Folsom, CA, USA). A Hewlett-Packard 7683 autosampler was used to carry out automated split injection. The FID temperature was set at 280 °C, and injection occurred at 250 °C. The oven temperature program that was employed was: hold at 40 °C for one minute; heat at 40 °C to 175 °C at a rate of 25 °C per minute; heat at 175 °C to 250 °C at a rate of 4 °C per minute; and hold at 250 °C for ten minutes. The column head pressure was kept constant and set at 32.44 psi. For all injections, a split injection liner (4 mm i.d.; Agilent, Santa Clara, CA, USA) was used. The flow rates of gases were helium (carrier), 2.1 mL min$^{-1}$; helium (make up), 30.0 mL min$^{-1}$; compressed air, 350 mL min$^{-1}$; and hydrogen, 45 mL min$^{-1}$.

In order to determine the Atherogenicity Index (AI), the Equation (2) was used based on the percentages of the fatty acids C12:0, C14:0, and C16:0:

$$\text{AI} = \frac{\text{C12}:0 + 4 \times \text{C14}:0 + \text{C16}:0}{(\text{PUFA} + \text{MUFA})} \tag{2}$$

where, PUFA and MUFA denote the polyunsaturated and monounsaturated fatty acids, respectively.

### 2.7. Total Phenolic Content

Total Phenols of cheese samples were extracted by methanol according to the method of Giorgio et al. [21]. Total phenolic content (TPC) was determined by Folin Ciocalteu's method [22]. The cheese's total phenolic content was expressed as mgr of gallic acid equivalents (GAE) per 100 gr of the sample.

### 2.8. Volatile Organic Compounds of Forage and Cheese

A modified headspace technique was used to examine the terpenoid compounds present in the cheese and forage samples [23]. The detected VOCs were identified using their MS spectra. The amount of volatile compounds present in each cheese sample was expressed in arbitrary units AU (peak area 9 × 10$^{-3}$).

### 2.9. Statistical Analysis

The statistical software Jeffreys's Amazing Statistics Program JASP (JASP v 0.14. https://jasp-stats.org/download/ (accessed on 20 January 2020)) was used to analyze the data [24]. The Student's *t*-test was used to compare the physical and chemical characteristics, fatty acid profile, and MDA concentration of the cheese from the plains and the mountain (5% level of significance). When the distribution was non-normal, the non-parametric tests Kruskal–Wallis and Mann–Whitney were used to compare the data.

## 3. Results and Discussion

### 3.1. Floristic Composition

The percentage of each functional group in the floristic composition of the three mountainous grasslands is presented in Figure 1.

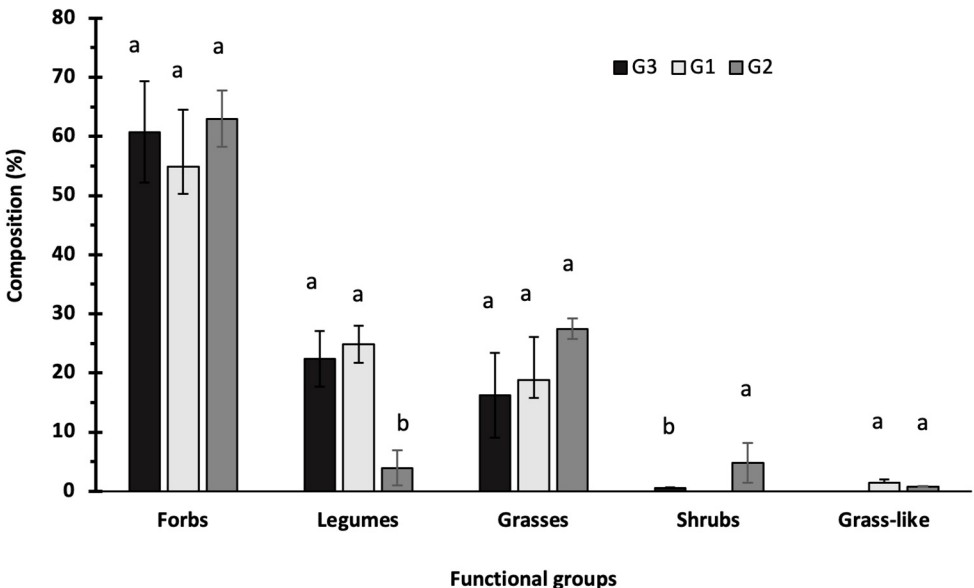

**Figure 1.** Floristic composition in functional groups of the studied mountainous grasslands G1, G2 and G3. Means ± SD for the same parameter followed by the same letter are not significantly different ($p \leq 0.05$).

Forbs was the major functional group in the three grasslands G1, G2, and G3, and there was no significant ($p > 0.05$) difference among them (Figure 1). The grasses' participation in the studied grasslands did not differ significantly between them. However, the contribution of legumes in G2 was significantly lower in comparison with the two other grasslands in which there was no significant difference. On the contrary, woody species were recorded in both G2 and G3 grasslands at Aetia. Climate and environmental factors affect species composition and facilitate the establishment of a number of species in a variety of habitats [25–27].

### 3.2. Volatile Organic Compound Content of the Pasture

The Volatile Organic Compounds (VOCs) identified in each functional group are presented in Table 1 and Figure 2. The most abundant VOCs were aldehydes, alcohols, and terpenes. Similar results were reported by Povolo et al. [28] in the forage of lowland pasture in Italy and by Bozoudi et al. [29] in mountainous regions in Greece. The volatile fraction of grasslands' forage was found to be rich in alcohols, such as 1-hexanol and 3,4-Hexanediol, 3,4-bis(4-hydroxyphenyl). Similar findings in relation to alcohol profiles have been reported by Povolo et al. [28]. Additionally, the forbs that were more widespread throughout all grasslands are rich in terpenes, which could be transferred to milk [30].

**Table 1.** Volatile Organic Compound (VOC) profiles of different functional groups (Forbs F, Legumes L, Grasses G) from the mountainous grasslands (G1, G2, G3).

| Volatile Organic Compounds Profile (VOCs) [1] | FG2 | LG2 | LG1 | FG1 | LG3 | FG3 | GG1 | LG3 | GG3 |
|---|---|---|---|---|---|---|---|---|---|
| Ketones | | | | | | | | | |
| Alcohols | | | | | | | | | |
| 1-Hexanol | 3.4 | 2.4 | 7.5 | 1.4 | n.d. | 3.6 | 5.9 | 2.7 | 3.2 |
| 2-Hexenol | 10.2 | n.d. | n.d. | n.d. | 10.5 | n.d. | n.d. | 5.1 | 8.1 |
| 3,4-Hexanediol, 3,4-bis(4-hydroxyphenyl) | 12.7 | 12.7 | 4.5 | 1.9 | 4.9 | 4.0 | 10.4 | 7.3 | 12.4 |
| 1-undecanol | n.d. | n.d. | n.d. | n.d. | 4.2 | n.d. | n.d. | n.d. | n.d. |
| Phenylethyl alcohol | n.d. | n.d. | n.d. | n.d. | 3 | 2.0 | n.d. | 1.3 | n.d. |

**Table 1.** *Cont.*

| Volatile Organic Compounds Profile (VOCs) [1] | Functional Groups | | | | | | | | |
|---|---|---|---|---|---|---|---|---|---|
| | **FG2** | **LG2** | **LG1** | **FG1** | **LG3** | **FG3** | **GG1** | **LG3** | **GG3** |
| Esters | | | | | | | | | |
| Aldehydes | | | | | | | | | |
| Benzaldehyde | n.d. | n.d. | n.d. | 3.7 | 4.4 | 2.6 | n.d. | 2.5 | 7.9 |
| Benzaldehyde, 2-hydroxy | 6.8 | n.d. | n.d. | n.d. | n.d. | n.d. | n.d. | n.d. | n.d. |
| Trichlorobenzaldehyde | n.d. | 5.8 | n.d. | n.d. | n.d. | n.d. | n.d. | n.d. | n.d. |
| 2,3,6-trichlorobenzaldehyde | n.d. | n.d. | n.d. | n.d. | n.d. | n.d. | n.d. | 3.9 | n.d. |
| Hydrocarbons | | | | | | | | | |
| 1,3-Budadiene, 2,3-dimethyl | 5.3 | n.d. | n.d. | 1.1 | n.d. | 3.1 | 4.2 | n.d. | 2.1 |
| Benzene, 1-ethyl-2,4-dimethyl | 2.9 | n.d. | n.d. | n.d. | n.d. | n.d. | n.d. | n.d. | n.d. |
| Cyclohexane, 1,2-dimethyl | n.d. | 10.9 | 33.9 | n.d. | n.d. | 8.2 | 11.0 | n.d. | n.d. |
| 1-tetradecene | n.d. | n.d. | n.d. | 23.2 | n.d. | n.d. | n.d. | n.d. | n.d. |
| 2-cyclopenten-1-one2-hydroxy-3-methyl | n.d. | n.d. | n.d. | 6.2 | n.d. | n.d. | n.d. | n.d. | n.d. |
| Acids | | | | | | | | | |
| Gibberellic acid | n.d. | n.d. | n.d. | n.d. | n.d. | 1.8 | n.d. | n.d. | n.d. |
| Dodecanoic acid | n.d. | n.d. | n.d. | n.d. | n.d. | n.d. | n.d. | 5.1 | n.d. |
| Terpenes | | | | | | | | | |
| tricyclene | 9.0 | n.d. | n.d. | n.d. | n.d. | n.d. | n.d. | n.d. | n.d. |
| b-pinene | 7.4 | n.d. | n.d. | 4.2 | n.d. | n.d. | n.d. | n.d. | n.d. |
| γ-terpinene | 4.3 | n.d. | n.d. | n.d. | n.d. | n.d. | n.d. | n.d. | n.d. |
| terpinolene | 6.2 | n.d. | n.d. | n.d. | n.d. | n.d. | n.d. | n.d. | n.d. |
| eucalyptol | n.d. | n.d. | n.d. | 4.3 | n.d. | 3.2 | n.d. | n.d. | n.d. |
| 3-carene | n.d. | n.d. | n.d. | n.d. | n.d. | 1.7 | n.d. | n.d. | n.d. |
| linalool | n.d. | n.d. | n.d. | n.d. | n.d. | 1.5 | n.d. | n.d. | n.d. |
| camphor | n.d. | n.d. | n.d. | n.d. | n.d. | 4.5 | n.d. | n.d. | n.d. |

[1] Results expressed in Arbitrary Units (AU = peak area $\times 10^{-3}$). n.d. not detected.

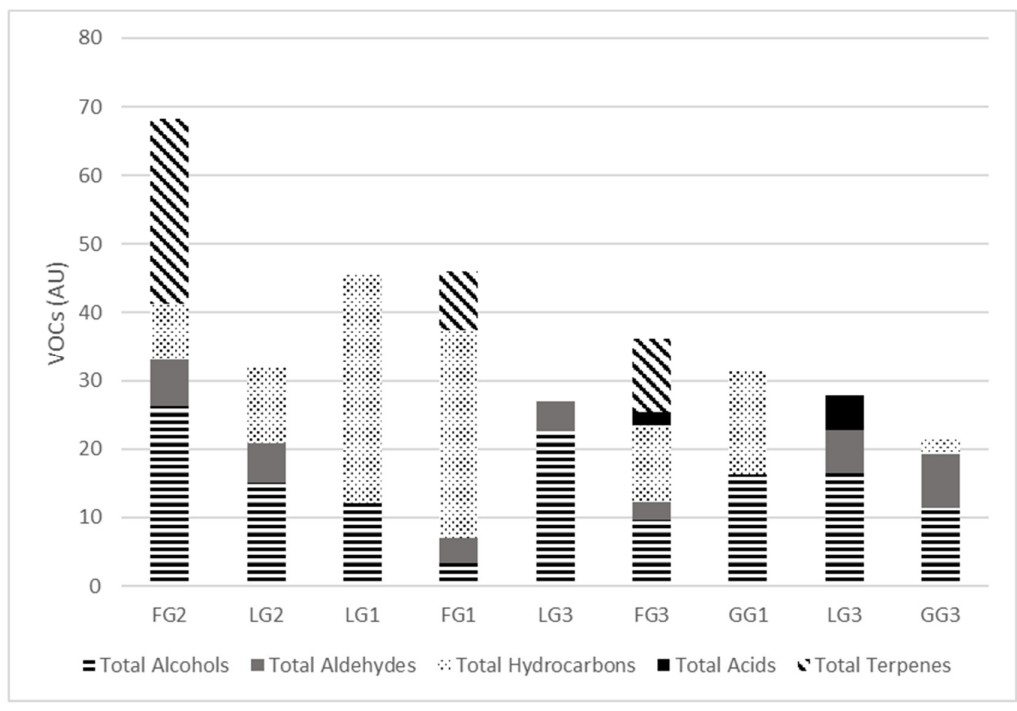

**Figure 2.** Volatile Organic Compound (VOC) profiles of different functional groups (Forbs F, Legumes L, Grasses G) from the mountainous grasslands (G1, G2, G3).

### 3.3. Physicochemical Characteristics of Cheese

Table 2 displays the compositional data of mature cheeses collected from the two regions in the same season, in two different years (Sampling A, B). Cheeses produced in Area M had a higher pH value than cheese produced in Area P in both samplings. Specifically, in Sampling A pH value of the low-land was significantly higher than the pH value of the mountainous area. According to the current regulations for feta cheese production in Greece, feta cheese has typically a pH of 4.4–4.6, a moisture content of about <56%, and a minimum fat in dry matter (FDM) of 43% [31,32]. All samples in the current study were within the permitted limits. The salt concentration seemed to be higher in the samples of the mountainous area in both samplings, with a significant difference ($p < 0.05$) in sampling B. The cheeses made in both areas had similar physicochemical characteristics, such as moisture, fat content and FDM (%), which is consistent with the results of Bozoudi et al. [33]. Carpino et al. [34] observed that feed had no discernible effect on the salt and fat of 7-month-old Ragusano cheese.

**Table 2.** Physicochemical characteristics of cheese manufactured in the two areas (Plains-Area P and Mountainous grasslands-Area M) (mean $\pm$ SD).

| Physicochemical Characteristics | Sampling A | | | Sampling B | | |
|---|---|---|---|---|---|---|
| | Area P | Area M | *p* Value | Area P | Area M | *p* Value |
| pH | 4.55 [a] $\pm$ 0.08 | 4.29 [b] $\pm$ 0.03 | 0.009 | 4.36 $\pm$ 0.13 | 4.16 $\pm$ 0.15 | 0.630 |
| Moisture (%) | 53.70 $\pm$ 1.10 | 51.16 $\pm$ 0.83 | 0.076 | 53.65 $\pm$ 0.93 | 52.30 $\pm$ 0.95 | 0.358 |
| Fat (%) | 27.75 $\pm$ 0.60 | 29.04 $\pm$ 0.90 | 0.243 | 26.28 $\pm$ 1.04 | 27.75 $\pm$ 0.92 | 0.315 |
| FDM (%) | 60.54 $\pm$ 0.36 | 59.42 $\pm$ 1.39 | 0.523 | 56.63 $\pm$ 0.66 | 58.50 $\pm$ 0.30 | 0.550 |
| Protein (%) | 15.70 $\pm$ 0.23 | 16.19 $\pm$ 0.34 | 0.265 | 16.21 [a] $\pm$ 0.40 | 18.47 [b] $\pm$ 0.30 | 0.039 |
| NaCl (%) | 1.63 $\pm$ 0.13 | 1.80 $\pm$ 0.13 | 0.394 | 1.15 [a] $\pm$ 0.04 | 2.09 [b] $\pm$ 0.03 | 0.001 |

Means $\pm$ SD for the same parameter with different superscripts (in lower case) are significantly different at $p \leq 0.05$.

The highest average protein content was found in feta produced in Area M, which differed significantly from the feta produced in Area P in the second sampling.

### 3.4. Fatty Acid Composition

Cheese contains a range of bioactive molecules, the most significant of which are fatty acids, in addition to the necessary nutrients of calcium, protein, and vitamins. According to research, cheese fat has 400 distinct FAs, making it the most complex fat in the human diet [35]. The fatty acid (FA) profile of mature cheeses obtained from the two areas of the two samplings is shown in Table 3. Of all fatty acids determined, the five dominant fatty acids were the saturated capric (C10:0), myristic (C14:0), palmitic (C16:0), stearic (C18:0) and monounsaturated oleic (C18:1$\omega$9c) acid, in both samplings and types of cheese, which is consistent with previous research [36–38]. Palmitic acid (C16:0) was found to be the most abundant in all samples. In previous research, it was found to be predominant in cheeses like Parmigiano Reggiano [39], Roncal, Mahon [40], Swiss, Blue, or Roquefort [41] and feta cheese [42]. Butyric acid (C4:0) is an important fatty acid of feta cheese, contributing to its flavor that had no significant difference between the two types of cheese ($p \leq 0.05$) [43].

For both types of cheese, the GC analysis revealed that the majority of the fatty acids were saturated (SFAs), which is consistent with the findings of previous studies [36,44]. It is crucial to note that the amount of SFAs, which are known to be health risk factors, were significantly lower ($p \leq 0.05$) in both types of milk from the mountainous area (Table 4) as well as the final product from the same area (Table 3). The amounts of the SFAs ranged in a similar way to earlier research [36,37]. Monounsaturated fatty acids (MUFA) showed the opposite tendency and their values differed significantly in Sampling A ($p \leq 0.05$). Polyunsaturated fatty acids (PUFA) seemed to have a slight increase in cheeses from mountainous areas (Area M).

**Table 3.** Fatty acid composition (%) of cheese manufactured in the two areas (Plains-Area P, mountainous grasslands—Area M) (mean ± SD).

| Fatty Acid Esters | Sampling A | | | Sampling B | | |
|---|---|---|---|---|---|---|
| | Area P | Area M | *p* Value | Area P | Area M | *p* Value |
| C4:0 | 2.61 ± 0.13 | 2.58 ± 0.05 | NS | 2.64 ± 0.13 | 2.60 ± 0.06 | NS |
| C6:0 | 2.13 ± 0.04 | 1.94 ± 0.03 | * | 2.16 ± 0.17 | 2.04 ± 0.08 | NS |
| C8:0 | 2.16 ± 0.05 | 1.88 ± 0.02 | * | 2.10 ± 0.20 | 2.00 ± 0.11 | NS |
| C10:0 | 7.07 ± 0.18 | 5.85 ± 0.05 | * | 6.63 ± 0.70 | 6.40 ± 0.53 | * |
| C11:0 | n.d. | n.d. | - | 0.08 ± 0.03 | 0.04 ± 0.01 | NS |
| C12:0 | 4.29 ± 0.11 | 3.37 ± 0.02 | * | 3.72 ± 0.38 | 3.48 ± 0.23 | NS |
| C13:0 | 0.15 ± 0.03 | 0.15 ± 0.01 | NS | 0.15 ± 0.01 | 0.13 ± 0.02 | NS |
| C14:0 | 11.83 ± 0.24 | 10.29 ± 0.08 | * | 10.76 ± 0.38 | 10.67 ± 0.25 | NS |
| C14:1 | 0.33 ± 0.03 | 0.23 ± 0.02 | * | 0.23 ± 0.02 | 0.26 ± 0.04 | NS |
| C15:0 | 1.22 ± 0.03 | 1.09 ± 0.02 | * | 1.08 ± 0.02 | 1.08 ± 0.02 | NS |
| C15:1 | n.d. | n.d. | - | n.d. | n.d. | - |
| C16:0 | 28.28 ± 0.42 | 25.53 ± 0.19 | * | 26.92 ± 0.38 | 26.66 ± 0.27 | NS |
| C16:1 | 1.11 ± 0.03 | 0.87 ± 0.02 | * | 0.89 ± 0.06 | 0.94 ± 0.05 | NS |
| C17:0 | 0.84 ± 0.08 | 0.94 ± 0.06 | NS | 0.83 ± 0.05 | 0.88 ± 0.05 | NS |
| C17:1 | n.d. | n.d. | - | n.d. | n.d. | - |
| C18:0 | 9.32 ± 0.29 | 12.21 ± 0.16 | * | 11.52 ± 0.95 | 11.94 ± 0.75 | NS |
| C18:1t | 0.52 ± 0.06 | 0.52 ± 0.03 | NS | 0.57 ± 0.02 | 0.46 ± 0.03 | * |
| C18:1cn9 | 21.21 ± 0.50 | 24.17 ± 0.26 | * | 22.84 ± 1.00 | 24.19 ± 0.60 | * |
| C18:1n7 | 0.50 ± 0.04 | 0.50 ± 0.02 | NS | 0.52 ± 0.04 | 0.50 ± 0.04 | NS |
| C18:2t | 0.22 ± 0.04 | 0.44 ± 0.04 | * | 0.36 ± 0.03 | 0.39 ± 0.06 | NS |
| C18:2c | 2.83 ± 0.09 | 3.02 ± 0.06 | * | 2.73 ± 0.14 | 2.73 ± 0.10 | NS |
| C18:3γ | n.d. | n.d. | - | n.d. | n.d. | - |
| C18:3α | 1.27 ± 0.08 | 1.59 ± 0.05 | * | 0.96 ± 0.09 | 1.16 ± 0.06 | * |
| CLA | 0.85 ± 0.10 | 1.25 ± 0.06 | * | 0.90 ± 0.07 | 1.01 ± 0.07 | NS |
| C20:0 | 0.31 ± 0.04 | 0.46 ± 0.03 | * | 0.48 ± 0.05 | 0.51 ± 0.04 | NS |
| C20:1n9 | 0.04 ± 0.02 | 0.09 ± 0.03 | * | 0.03 ± 0.00 | 0.03 ± 0.00 | NS |
| C20:2 | 0.04 ± 0.01 | 0.01 ± 0.00 | * | n.d. | n.d. | NS |
| C21:0 | 0.08 ± 0.04 | 0.08 ± 0.03 | NS | 0.06 ± 0.02 | 0.10 ± 0.02 | NS |
| C20:3n6 | n.d. | n.d. | - | n.d. | n.d. | - |
| C20:4n6 | 0.38 ± 0.08 | 0.25 ± 0.04 | NS | 0.28 ± 0.03 | 0.18 ± 0.03 | NS |
| C20:3n3 | n.d. | n.d. | - | n.d. | n.d. | - |
| C22:0 | 0.10 ± 0.03 | 0.20 ± 0.05 | * | 0.18 ± 0.07 | 0.25 ± 0.08 | * |
| EPA | 0.04 ± 0.02 | 0.17 ± 0.05 | * | 0.11 ± 0.05 | 0.05 ± 0.01 | NS |
| C22:1n11 | n.d. | n.d. | - | n.d. | n.d. | - |
| C22:1n9 | n.d. | n.d. | - | n.d. | n.d. | - |
| C22:2 | n.d. | n.d. | - | n.d. | n.d. | - |
| C23:0 | n.d. | n.d. | - | 0.06 ± 0.02 | 0.05 ± 0.01 | NS |
| C22:4n6 | n.d. | n.d. | - | n.d. | n.d. | - |
| C24:0 | n.d. | n.d. | - | 0.03 ± 0.01 | 0.05 ± 0.02 | NS |
| C22:5n3 | 0.12 ± 0.04 | 0.25 ± 0.05 | * | 0.10 ± 0.04 | 0.18 ± 0.06 | * |
| C24:1 | n.d. | n.d. | - | n.d. | n.d. | - |
| DHA | 0.01 ± 0.00 | 0.03 ± 0.00 | * | 0.06 ± 0.02 | 0.04 ± 0.01 | NS |
| SFA | 70.54 ± 0.67 | 66.60 ± 0.33 | * | 69.40 ± 1.16 | 67.89 ± 0.56 | * |
| PUFA | 5.74 ± 0.21 | 6.99 ± 0.09 | * | 5.53 ± 0.16 | 5.73 ± 0.13 | NS |
| MUFA | 23.72 ± 0.50 | 26.41 ± 0.28 | * | 25.08 ± 1.00 | 25.38 ± 0.50 | NS |
| UFA | 29.46 ± 0.67 | 33.40 ± 0.33 | * | 30.60 ± 1.16 | 31.11 ± 0.56 | NS |
| n-6 | 3.46 ± 0.12 | 3.70 ± 0.06 | * | 3.37 ± 0.12 | 3.31 ± 0.16 | NS |
| n-3 | 1.43 ± 0.07 | 2.04 ± 0.07 | * | 1.25 ± 0.17 | 1.42 ± 0.06 | NS |
| n-6/n-3 | 2.54 ± 0.16 | 1.87 ± 0.09 | * | 3.00 ± 0.57 | 2.36 ± 0.19 | * |
| PUFA/SFA | 0.08 ± 0.00 | 0.103 ± 0.01 | * | 0.08 ± 0.01 | 0.08 ± 0.01 | NS |
| AI | 2.82 ± 0.08 | 2.12 ± 0.05 | * | 2.44 ± 0.18 | 2.35 ± 0.09 | * |

NS, not significant, * $p \leq 0.05$; n.d., not detected.

**Table 4.** Fatty acid composition (%) of milk from the two areas (Plains-Area P, mountainous grasslands—Area M) (mean ± SD).

| Fatty Acid Esters | Area P | Area M | *p* Value |
|:---:|:---:|:---:|:---:|
| C4:0 | 2.21 ± 0.07 | 2.01 ± 0.08 | * |
| C6:0 | 2.00 ± 0.03 | 1.82 ± 0.08 | * |
| C8:0 | 2.00 ± 0.02 | 1.98 ± 0.11 | NS |
| C10:0 | 6.58 ± 0.08 | 6.53 ± 0.44 | NS |
| C11:0 | 0.08 ± 0.03 | 0.05 ± 0.01 | * |
| C12:0 | 4.27 ± 0.06 | 3.20 ± 0.10 | * |
| C13:0 | 0.11 ± 0.01 | 0.12 ± 0.01 | NS |
| C14:0 | 12.50 ± 0.8 | 9.56 ± 0.17 | * |
| C14:1 | 0.28 ± 0.03 | 0.18 ± 0.04 | * |
| C15:0 | 1.38 ± 0.02 | 1.05 ± 0.08 | * |
| C15:1 | n.d. | n.d. | - |
| C16:0 | 30.37 ± 0.18 | 26.00 ± 0.32 | * |
| C16:1 | 1.27 ± 0.03 | 0.73 ± 0.05 | * |
| C17:0 | 0.88 ± 0.02 | 0.96 ± 0.05 | NS |
| C17:1 | n.d. | n.d. | - |
| C18:0 | 8.92 ± 0.15 | 13.91 ± 0.35 | * |
| C18:1t | 0.44 ± 0.01 | 0.56 ± 0.03 | * |
| C18:1cn9 | 20.09 ± 0.26 | 22.42 ± 0.56 | * |
| C18:1n7 | 0.42 ± 0.01 | 0.38 ± 0.02 | NS |
| C18:2t | 0.20 ± 0.06 | 0.34 ± 0.05 | NS |
| C18:2c | 2.65 ± 0.05 | 3.04 ± 0.17 | * |
| C18:3γ | n.d. | n.d. | - |
| C18:3α | 1.16 ± 0.02 | 1.86 ± 0.09 | * |
| CLA | 0.71 ± 0.02 | 1.10 ± 0.09 | * |
| C18:4n3 | n.d. | n.d. | - |
| C20:0 | 0.29 ± 0.01 | 0.47 ± 0.04 | * |
| C20:1n9 | n.d. | n.d. | - |
| C20:2 | n.d. | n.d. | - |
| C21:0 | 0.11 ± 0.01 | 0.10 ± 0.01 | NS |
| C20:3n6 | n.d. | n.d. | - |
| C20:4n6 | 0.23 ± 0.01 | 0.29 ± 0.06 | NS |
| C20:3n3 | n.d. | n.d. | - |
| C22:0 | 0.17 ± 0.01 | 0.23 ± 0.02 | * |
| EPA | 0.12 ± 0.02 | 0.19 ± 0.03 | NS |
| C22:1n11 | n.d. | n.d. | - |
| C22:1n9 | n.d. | n.d. | - |
| C22:2 | n.d. | n.d. | - |
| C23:0 | 0.08 ± 0.03 | 0.21 ± 0.08 | NS |
| C22:4n6 | 0.02 ± 0.00 | 0.06 ± 0.01 | * |
| C24:0 | 0.08 ± 0.02 | 0.14 ± 0.02 | NS |
| C22:5n3 | 0.18 ± 0.01 | 0.32 ± 0.07 | * |
| C24:1 | n.d. | n.d. | - |
| DHA | 0.02 ± 0.01 | 0.08 ± 0.01 | * |
| SFA | 72.05 ± 0.25 | 68.33 ± 0.88 | * |
| PUFA | 5.39 ± 0.03 | 7.31 ± 0.38 | * |
| MUFA | 22.55 ± 0.59 | 24.35 ± 3.34 | * |
| UFA | 27.95 ± 0.25 | 31.65 ± 0.88 | * |
| n-6 | 3.19 ± 0.03 | 3.76 ± 0.21 | * |
| n-3 | 1.49 ± 0.03 | 2.45 ± 0.15 | * |
| n-6/n-3 | 2.15 ± 0.06 | 1.58 ± 0.06 | * |
| PUFA/SFA | 0.07 ± 0.01 | 0.11 ± 0.01 | * |
| AI | 3.03 ± 0.04 | 2.28 ± 0.07 | * |

NS, not significant, * $p \leq 0.05$; n.d., not detected.

Research on the impact of pasture elevation on milk fat-conjugated linoleic acid (CLA) levels found a correlation with the particular botanical composition of pastures [11,45]. Indeed, it has been noted that lowland pasture or silage feeding has a detrimental impact

on the CLA content of fat milk, and under these feeding circumstances, CLA values lower than 0.6% can be anticipated [46,47]. As shown in Table 3, CLA values showed an increase in milk from Area M that differed significantly from the Area P group, a fact that seems to have an impact on cheese. As presented in Figure 3, CLA amounts were higher in the cheeses from mountainous area and they were significantly different in Sampling A. CLA levels was very similar with those in other cheeses that were manufactured in mountainous areas [7]. Numerous health advantages of CLA, especially its major isomer, cis-9, trans-11 C18:2, have been demonstrated in studies, including anticarcinogenic, antiatherogenic, immune system boosting, and antidiabetic properties [48].

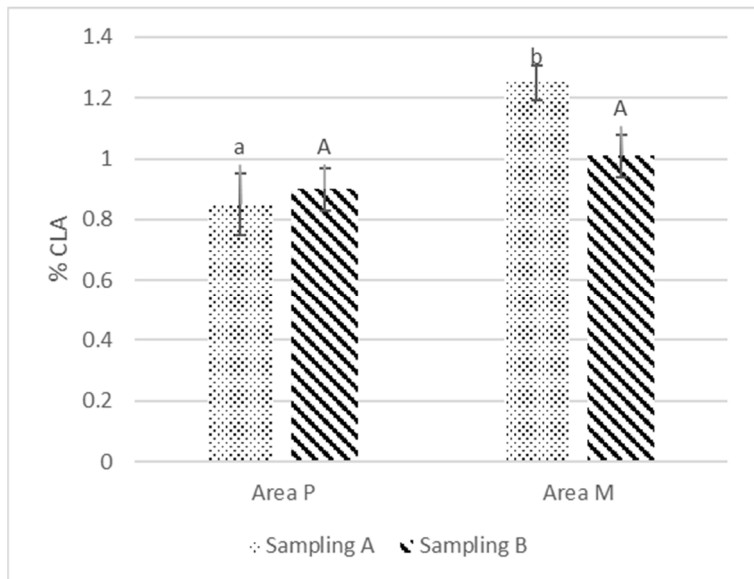

**Figure 3.** Conjugated Linoleic Acid—CLA values (%) for both cheese types (Area P and Area M) from two different samplings. Means ± SD for the same parameter with different superscripts (in upper case or lower case) are significantly different at $p \leq 0.05$.

The Atherogenicity Index (AI) was determined using the values that are shown in Table 3 and Equation (2). AI examines the correlation between the total sum of saturated fatty acids and the total sum of unsaturated fatty acids, with the former being regarded as promoting the adhesion of lipids to cells in the immune and circulatory systems (pro-atherogenic), while the latter is considered to inhibit plaque aggregation and reduce levels of esterified fatty acids, cholesterol, and phospholipids, thereby preventing the occurrence of coronary diseases on both small and large scales (antiatherogenic) [49]. AI also takes into consideration the diverse impacts that individual fatty acids may have on human health, specifically their potential to increase the likelihood of pathogenic events such as the development of atheroma or the formation of blood clots. According to the values reported in Tables 3 and 4, the AI value is lower in the cheese and milk from Area M ($p \leq 0.05$).

The composition of fatty acids in dairy products, particularly in cheese, is related to milk composition. Additionally, milk fat is directly influenced by dairy farming practices, indicating that the nutritional control of factors such as forage type, concentrate ratio, and the incorporation of feeds containing high levels of polyunsaturated fatty acids can shape the fatty acid profile of milk [50]. The comparison between the FA profile of milk of herds grazing on the plains and in mountainous grasslands is presented in Table 4. There were many similarities between the FA profile of milk and cheese. However, statistical differences were observed in more fatty acids between the milk of the two areas. Specifically, the four dominant FAs in milk were the same as in cheese, i.e., C14:0, C16:0, C18:0, and C18:1c, while C16:0 (palmitic acid) was the most abundant. All of these FAs were significantly influenced by the altitude. Specifically, C14:0 and C16:0 were significantly lower in the mountainous area, while C18:0 and C18:1c showed an increase ($p \leq 0.05$).

Considering the unsaturated FA of milk, the concentrations of C18:1t ($p \leq 0.01$), C18:1cn9 ($p \leq 0.05$), CLA ($p \leq 0.05$), C18:2c ($p \leq 0.05$), C18:3$\alpha$ ($p \leq 0.05$), and C22:4n6 ($p \leq 0.05$) were higher in milk from the mountain group. In previous research [44,51], it has been observed that increasing the fresh herbage intake by animals also led to increased levels of C18:1t and CLA.

Monounsaturated fatty acid (MUFAs) levels were found to be higher ($p \leq 0.05$) in milk from the mountain group. MUFAs have been very effective at reducing the risk of coronary heart disease due to their ability to decrease blood cholesterol, particularly DHA [49]. DHA levels were also found to be higher in the same group. Nevertheless, this difference was not observed in both samplings in cheese manufacture.

Animal nutrition has a significant impact on the fatty acid profile of raw milk, with significant differences among most of the groups and especially between FAs groups that have a major influence on human health [52]. Most of these differences were also observed between the cheeses that were manufactured in areas with different altitudes, but not all of them.

*3.5. Lipid Oxidation*

The following table (Table 5) presents the extent of lipid oxidation in the cheese samples. Lipid oxidation values exhibit a notable increase over the course of eight months. Furthermore, when comparing the lipid oxidation values observed after ten months of ripening with those from the initial two months, a statistically significant difference is observed. The two types of cheese seemed to have similar amounts of MDA that were not statistically different. Lipid oxidation is a noteworthy quality concern in processed dairy products, especially when stored. The results of our investigation indicate that lipid oxidation does not significantly occur during ripening, and the values are consistent with those of other studies [38,53]. Moreover, the MDA values of the cheeses of the mountainous grasslands (Area M) are slightly lower in the tenth month of ripening.

**Table 5.** Lipid oxidation (ng MDA g$^{-1}$) changes during ripening for cheese from the two areas (Plains-Area P, Mountainous grasslands—Area M).

| | MDA (ng g$^{-1}$) | |
| --- | --- | --- |
| | **2 Months** | **10 Months** |
| Area P | 26.67 $\pm$ 4.31 [a] | 128.37 $\pm$ 11.88 [b] |
| Area M | 54.19 $\pm$ 8.47 [a] | 114.88 $\pm$ 8.51 [b] |

Mean for the same parameter followed by the same letter is not significantly different ($p \leq 0.05$).

*3.6. Total Phenolic Content*

The TPC of either the control cheeses or cheeses from mountainous areas measured over the ripening period is presented in Figure 4. As demonstrated, the samples from the mountainous grasslands (Area M) have slightly higher values of TPC than the cheeses from the plains (Area P). The TPC of cheese from Area P increased over time, which is consistent with previous studies [54–56]. The TPC of the cheeses of Area M seems to increase during ripening, whereas at 60 days, it showed a slight decrease.

Phenolic compounds, a category of phytochemicals with potential health benefits, are widely distributed in plants [57]. The presence of these compounds in milk and dairy products is linked to the consumption of certain forage species, natural pasture, aromatic plants, and free-range grazing [58]. The TPC of forage or pasture depends on botanical species, environmental factors, and harvest or intake stage [59]. In our research, higher TPC values were observed in the samples from Area M, which is due to the different diets of the animals in these two areas.

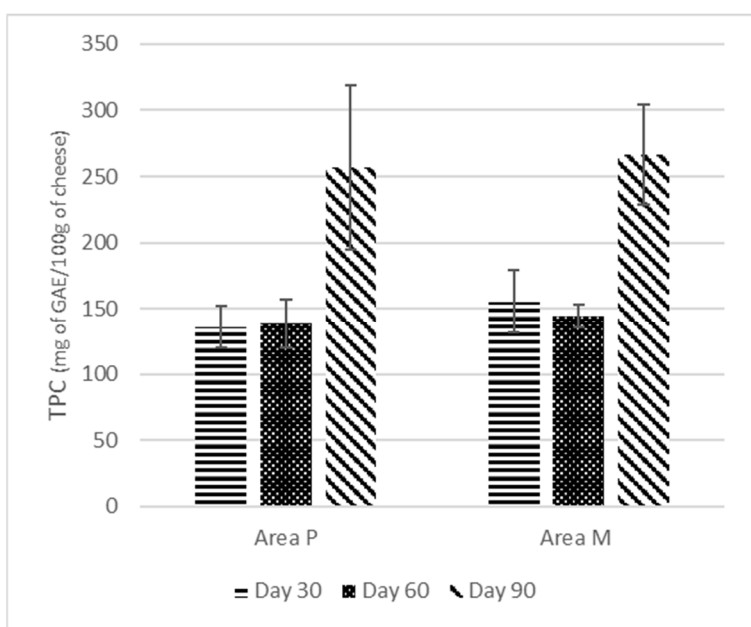

**Figure 4.** Total phenolic content (TPC) (mg of GAE/100 g of cheese) during ripening for cheese from the two areas (Plains-Area P, Mountainous grasslands—Area M).

### 3.7. Volatile Organic Compound Content of the Cheeses

Different types of volatile compounds in samples of feta cheese might be identified according to the SPME/GC-MS analysis presented in Table 6. A total of 34 volatile compounds were identified and classified into the following chemical groups: alcohols, ketones, aldehydes, acids, hydrocarbons, esters and terpenoids. The most abundant groups in cheese samples were esters, hydrocarbons and alcohols with esters made from milk from the flocks that grazed in plains (Area P) and hydrocarbons in the cheese samples made from milk from flocks that grazed in mountainous grasslands (Area M).

In the analysis of feta cheese samples, several esters were identified, including 1-butanol-3-methyl-acetate, acetic acid ethyl ester, propyl acetate, butanoic acid pentyl ester, butanoic acid ethyl ester, orthoacetic acid trimethyl ester, 1-hexane, 4,5-dimethyl, furan, and 2-butyl tetrahydro. Notably, the cheese samples from Area P exhibited the highest abundance of esters. Esters can be formed through enzymatic or chemical processes involving the reaction of short-to medium-chain fatty acids with primary and secondary alcohols. They can also result from the transesterification of partial glycerides to ethanol, as documented in previous research [60,61]. The majority of cheese esters that are observed are described as having sweet, fruity, and flowery characteristics. Ethyl esters are especially known for their crucial role in the formation of a fruity character in cheese [62].

The hydrocarbons identified in the samples were pentane 2,3-dimethyl, propane,2-methyl,1-nitro, 1-hexane, 4,5-dimethyl, tetradecane, heptanal, decane, octane.2,2,7,7-tetramethyl. Hydrocarbons can originate both from feed and lipid oxidation [54].

Two alcohols (2 propanol,1-methoxy, 2 propanol,1-butoxy) were found to be in feta samples, in great amounts. The branched chain alcohol 3-methyl butanol, which comes from leucine via Strecker degradation, lends cheeses an alcoholic and flowery note [63,64].

Butanoic, hexanoic and thioacetic acid were found mainly in cheeses from the Area M. Due to their low sensory thresholds and distinctive flavor characteristics, the short- and medium-chain fatty acids present in feta cheese contribute to the background flavor of many cheese varieties [65].

**Table 6.** VOC profiles in mature feta cheese produced in two different areas (Plains-Area P, Mountainous grasslands—Area M).

| Volatile Organic Compounds (VOCs) [1] | Area P | | | | Area M | | | | | |
|---|---|---|---|---|---|---|---|---|---|---|
| | P1 | P2 | P3 | P4 | M1 | M2 | M3 | M4 | M5 | M6 |
| Total Ketones | | | | | | | | | | |
| Acetone | n.d. | n.d. | n.d. | n.d. | n.d. | n.d. | n.d. | n.d. | n.d. | n.d. |
| 2-Butanone | 0.9 | 5.1 | n.d. | n.d. | n.d. | n.d. | n.d. | 1.0 | n.d. | n.d. |
| 2-pentanone | n.d. | n.d. | n.d. | n.d. | n.d. | n.d. | n.d. | n.d. | n.d. | n.d. |
| 2-heptanone | n.d. | n.d. | n.d. | n.d. | n.d. | n.d. | n.d. | n.d. | n.d. | n.d. |
| Alcohols | | | | | | | | | | |
| 2-propanol | n.d. | n.d. | n.d. | n.d. | n.d. | n.d. | n.d. | n.d. | n.d. | n.d. |
| 1-Butanol, 3-methyl | n.d. | n.d. | n.d. | n.d. | n.d. | n.d. | n.d. | n.d. | n.d. | n.d. |
| 2 propanol,1-methoxy | 0.4 | 0.8 | 0.5 | 8.2 | 9.3 | 1.2 | 2.1 | 2.1 | n.d. | 23.9 |
| 2 propanol,1-butoxy | 4.2 | 43.6 | 8.6 | n.d. | 7.5 | 4.3 | 20.1 | 20.9 | n.d. | n.d. |
| Total Esters | | | | | | | | | | |
| Ethyl acetate | n.d. | n.d. | n.d. | n.d. | n.d. | n.d. | n.d. | 2.6 | n.d. | n.d. |
| Propyl acetate | n.d. | n.d. | 0.9 | n.d. | 3.1 | 1,6 | 1.8 | n.d. | 0.8 | n.d. |
| Butanoic acid, pentyl ester | n.d. | 32.4 | 10.0 | n.d. | n.d. | 5.3 | 19.2 | n.d. | 13.1 | n.d. |
| Hexanoic acid, ethyl ester | n.d. | n.d. | 6.1 | n.d. | 1.0 | 1.0 | 1.4 | n.d. | - | n.d. |
| Orthoacetic acid, trimethyl ester | n.d. | n.d. | 4.9 | 19.8 | 14.7 | 3.1 | 6.1 | n.d. | 40.3 | n.d. |
| Butanoic acid, ethyl ester | n.d. | n.d. | 8.5 | n.d. | 3.3 | n.d. | 3.2 | n.d. | n.d. | 4.3 |
| 1-butanol, 3-methyl acetate | n.d. | n.d. | 3.0 | 1.0 | n.d. | 1.9 | n.d. | n.d. | n.d. | 1.4 |
| 1-hexane, 4,5-dimethyl | n.d. | n.d. | 35.9 | n.d. | n.d. | n.d. | n.d. | n.d. | n.d. | n.d. |
| Furan, 2-butyl tetrahydro | 75.1 | n.d. | n.d. | n.d. | n.d. | n.d. | n.d. | n.d. | n.d. | n.d. |
| Total Aldeydes | | | | | | | | | | |
| Butanal, 3- methyl | n.d. | n.d. | n.d. | n.d. | n.d. | n.d. | n.d. | n.d. | 1.5 | n.d. |
| Total Hydrocarbons | | | | | | | | | | |
| Pentane | n.d. | n.d. | n.d. | n.d. | n.d. | n.d. | n.d. | n.d. | n.d. | n.d. |
| Pentane 2-methyl | n.d. | n.d. | n.d. | n.d. | n.d. | n.d. | n.d. | n.d. | n.d. | n.d. |
| Pentane 3-methyl | n.d. | n.d. | n.d. | n.d. | n.d. | n.d. | n.d. | n.d. | n.d. | n.d. |
| Pentane 2,3-dimethyl | n.d. | n.d. | 1.1 | 1.7 | 2.4 | 0.7 | n.d. | n.d. | 0.4 | 1.4 |
| Propane,2-methyl,1-nitro | 5.1 | 9.7 | n.d. | n.d. | n.d. | n.d. | n.d. | n.d. | n.d. | n.d. |
| 1-hexane, 4,5-dimethyl | 3.0 | 6.3 | 38.6 | n.d. | 39.7 | 37.1 | 30.3 | 12.8 | 24.3 | 43.7 |
| tetradecane | 0.4 | n.d. | 3.2 | n.d. | 1.5 | 3.5 | 1.7 | 4.1 | n.d. | n.d. |
| heptanal | n.d. | n.d. | n.d. | n.d. | n.d. | 1.9 | n.d. | 1.4 | n.d. | n.d. |
| Decane | n.d. | n.d. | n.d. | n.d. | n.d. | 2.9 | n.d. | n.d. | n.d. | n.d. |
| cyclohexane | n.d. | n.d. | n.d. | n.d. | n.d. | n.d. | n.d. | n.d. | n.d. | n.d. |
| Octane.2,2,7,7-tetramethyl | n.d. | n.d. | n.d. | 2.2 | n.d. | n.d. | n.d. | n.d. | 0.4 | n.d. |
| **Total Acids** | | | | | | | | | | |
| L-Lactic acid | n.d. | n.d. | n.d. | n.d. | n.d. | n.d. | n.d. | n.d. | n.d. | n.d. |
| Butanoic acid | n.d. | n.d. | n.d. | n.d. | 1.8 | n.d. | n.d. | 4.2 | n.d. | 3.7 |
| Hexanoic acid | n.d. | n.d. | n.d. | n.d. | n.d. | 1.8 | n.d. | n.d. | n.d. | n.d. |
| Thioacetic acid | n.d. | n.d. | n.d. | n.d. | n.d. | n.d. | 2.5 | n.d. | n.d. | 2.3 |
| **Total Terpenoids** | | | | | | | | | | |
| a-pinene | n.d. | n.d. | 1.5 | 1.5 | 1.2 | 1.7 | 1.6 | 2.3 | n.d. | n.d. |

[1] Results expressed in Arbitrary Units (AU = peak area $\times 10^{-3}$). n.d., not detected.

Moreover, a-pinene was identified in larger amounts in cheese samples from Area M. These terpenes found in cheese are likely derived from the plants that constitute the grazing animals' feed, a phenomenon that has been observed in artisanal cheeses produced in alpine settings [11,66]. These aromatic compounds are initially absorbed by the animals through their consumption of diverse pasture vegetation, subsequently becoming integrated into the milk and ultimately manifesting within the cheese matrix. Notably, Bosset et al. observed that alpine cheeses consistently exhibited elevated terpene levels compared to their plain counterparts [67].

The VOC profiles of individual samples from Area P (P) and Area M (M) are presented in Figure 5.

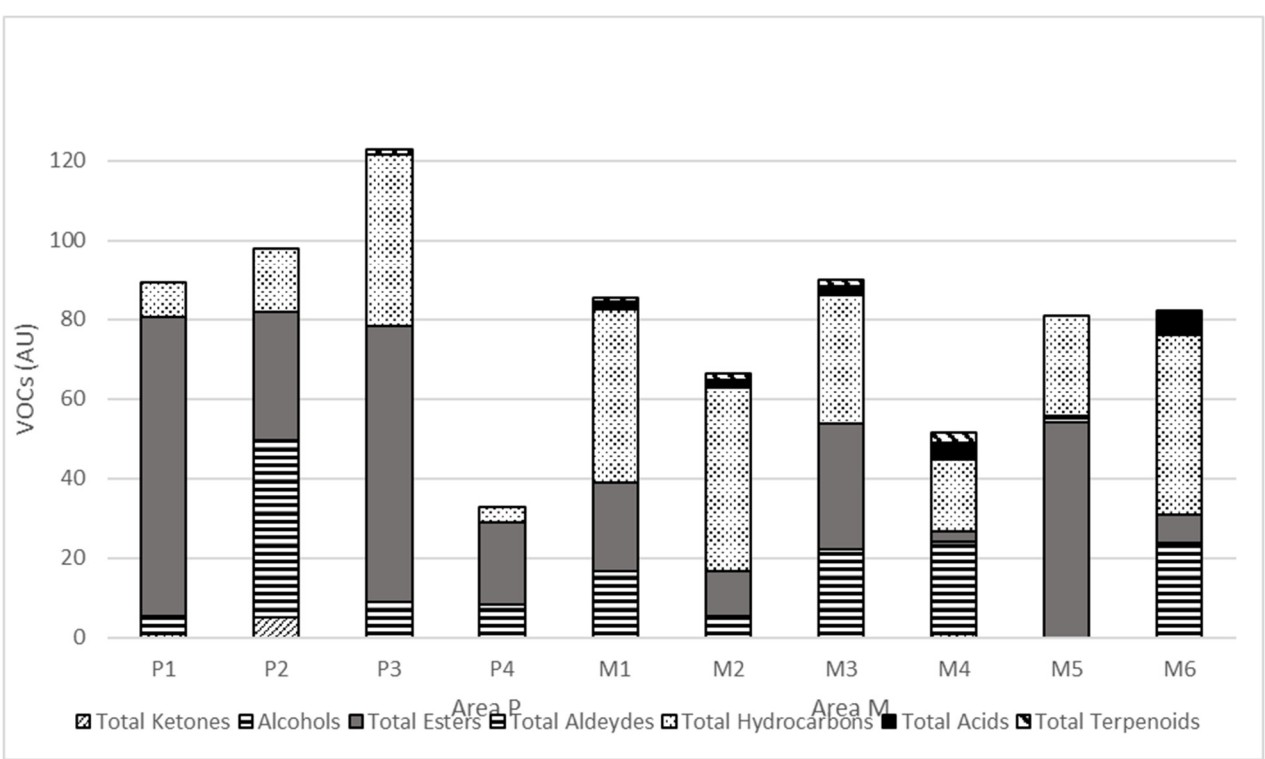

**Figure 5.** Volatile Organic Compound (VOC) profiles of different cheese samples from the two areas. (Plains-Area P, Mountainous grasslands—Area M).

It seems that compounds only found in cheeses from each distinct region might be regarded as distinguishing characteristics of regional cheeses. Terpenoids and total acids were detected in higher concentrations in the cheeses from mountainous regions.

## 4. Conclusions

Traditional feta cheeses have been produced at both low and high altitudes, and the chemical and physical characteristics, which make up the majority of their composition have been proven to be comparable. Additionally, the levels of MDA (malondialdehyde) and total phenolic content were comparable in both cheeses, but increased significantly during the ripening process. Moreover, according to our research, feeding the animals at different altitudes can significantly affect the milk's fatty acid composition. The cheese made from the milk of sheep and goats that grazed in the mountainous areas contains more conjugated linoleic acid (CLA) and fewer saturated fatty acids (SFAs). These results indicate that the pastures in the mountainous region, where the sheep and goats grazed, contribute positively to the nutritional quality of feta cheese. The volatile organic compound (VOC) profiles of the cheeses also exhibited differences among the two types of cheese, with esters, hydrocarbons, and alcohols being the most abundant VOC groups. While there were certain substances present in all samples that characterize traditional feta cheese, there were also some that were unique to cheeses from each specific area.

**Author Contributions:** Conceptualization, G.S.; methodology, M.M. and M.D.I.; software, M.M., M.D.I., M.K., Z.M.P. and E.M.A.; validation, M.M. and M.D.I.; formal analysis, M.M., M.D.I., Z.M.P., E.M.A. and M.K.; investigation, Z.M.P., E.M.A., M.K. and G.S.; resources, G.S.; data curation, Z.M.P., E.M.A., M.D.I. and M.K.; writing—original draft preparation, M.M. and M.D.I.; writing—review and editing, M.M., M.D.I., Z.M.P., E.M.A., M.K. and G.S.; visualization, M.M. and M.D.I.; supervision, G.S.; project administration, G.S.; funding acquisition, G.S. All authors have read and agreed to the published version of the manuscript.

**Funding:** This research was funded by the General Secretariat for Research and Innovation, Greece, grant number T1EDK-04475.

**Institutional Review Board Statement:** Not applicable.

**Informed Consent Statement:** Not applicable.

**Data Availability Statement:** The data presented in this study are available on request from the corresponding author.

**Conflicts of Interest:** The authors declare no conflict of interest.

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
