# Peer review of "Compositional Characteristics, Fatty Acid Profile, Phenolic Content and Volatile Organic Compounds (VOCs) of Feta Cheese Made in Mountainous Grasslands and Plains of Greece"

_2624-862X, doi:10.3390/dairy4040046_

Round 1
Reviewer 1 Report
Comments and Suggestions for Authors
This article is interesting, however some observations and suggestions are made.
Figure 1. Please put the standard deviations in each column of the data.
Table 1. In what units is VOCs reported? Please indicate in the table.
Table 2. What does M and P mean? Please explain in the footer.
Figure 2. Please put the description on the y-axis and the letters of the statistic to see significant differences.
Figure 3. Please put the letters of the statistic to see significant differences. Please remove the decimals in the y-axis.
Table 6. There should be no blank spaces in the tables, perhaps you can put a hyphen "-".
Figure 4. Please describe the y-axis, put the units.
The authors present physicochemical characterizations and compounds such as fatty acids and phenolic compounds. However, this article completely lacks discussion, they only make a very superficial comparison between cheeses, regions and materials that can influence cheeses. The structural discussion with the characteristics of the cheese, sensory and physical aspects, etc. is not observed. The integrated results are not observed regarding the discussion. The authors need to reinforce a lot in this. In addition to seeing other publications on the subject.
Author Response
Figure 1. Please put the standard deviations in each column of the data.
It has been done as requested
Table 1. In what units is VOCs reported? Please indicate in the table.
Thank you for your comment. Results expressed in Arbitrary Units (AU = peakarea Χ 10 -3 ). We added at the table.
Table 2. What does M and P mean? Please explain in the footer.
Explanation for Area M and Area P is added in all tables.
Figure 2. Please put the description on the y-axis and the letters of the statistic to see significant differences.
It has been done as requested.
Figure 3. Please put the letters of the statistic to see significant differences. Please remove the decimals in the y-axis.
We put the description on the y axis. As regards the phenolic content, no statistical analysis was carried out
Table 6. There should be no blank spaces in the tables, perhaps you can put a hyphen "-".
It has been done as requested. We corrected at the tables.
Figure 4. Please describe the y-axis, put the units.
It has been done as requested.
The authors present physicochemical characterizations and compounds such as fatty acids and phenolic compounds. However, this article completely lacks discussion, they only make a very superficial comparison between cheeses, regions and materials that can influence cheeses. The structural discussion with the characteristics of the cheese, sensory and physical aspects, etc. is not observed. The integrated results are not observed regarding the discussion. The authors need to reinforce a lot in this. In addition to seeing other publications on the subject.
Thank you for your comment. The discussion part was enhanced mainly in the phenolic content part. Also, some other publications were added.
Reviewer 2 Report
Comments and Suggestions for Authors
In this article, the authors present a comparative study on Feta cheese production in the mountainous grasslands and plains of Greece, unveiling noteworthy compositional variations. Cheeses from mountainous regions exhibit higher moisture, lower fat, and elevated protein content, along with a distinctive profile of volatile organic compounds (VOCs). The research highlights the favorable influence of grazing environments on the quality of Feta cheese, emphasizing the pivotal role of mountainous pastures in enhancing its nutritional characteristics.
The manuscript is written well and the discussion is easy to follow. However, the authors are encouraged to consider the following comments for further enhancing the manuscript.
For these reasons, I recommend accepting this manuscript for publication in “Dairy” after incorporating the comments.
1. Is there a specific reason for the presence of a full stop in the title? This isn't a common practice and I'm curious about the rationale behind it.
2. Could the data from Table 1 be represented in a figure similar to what the authors have done in Figure 4?
3. It might be beneficial to relocate Table 6 to the supporting information as Figure 4 presents the information more comprehensively. Comparing the data through a visual representation in a figure is clearer than reading the numbers in the table.
4. The equations in the text are not numbered, which may make it challenging for readers to follow. Numbering the equations could enhance readability.
5. Is it possible to ensure uniformity in the font size in Figure 1, specifically addressing the discrepancy at line 207? Additionally, could the authors provide an explanation for the representations 'a' and 'b' in Figure 1, and clarify the meaning of 'P≤0.05'?
6. Could the authors consider introducing abbreviations at their first instance in the manuscript, for instance, in Section 3.2, to ensure consistency and aid reader comprehension throughout the document?
7. In Table 1, it would be helpful for the authors to explicitly define each abbreviation, akin to the approach taken in other tables, such as 'n.d' at the end of the table.
8. Clarification on the meaning of 'p' and the p-values in Tables 2 and 3, respectively would be beneficial to avoid reader confusion and maintain consistency.
9. Regarding the caption of Figure 5 at line 343, could 'means' be adjusted to 'mean' for accuracy?
10. The manuscript's language must be reviewed to ensure it caters to a broader readership.
Comments on the Quality of English Language
The manuscript's language must be reviewed to ensure it caters to a broader readership.
Author Response
- Is there a specific reason for the presence of a full stop in the title? This isn't a common practice and I'm curious about the rationale behind it.
It was not something specific about the full stop in the title. We removed it. - Could the data from Table 1 be represented in a figure similar to what the authors have done in Figure 4?
It has been done as requested. - It might be beneficial to relocate Table 6 to the supporting information as Figure 4 presents the information more comprehensively. Comparing the data through a visual representation in a figure is clearer than reading the numbers in the table.
A Figure was made also for Table 1. Do you think that it would be better to relocate both Tables? - The equations in the text are not numbered, which may make it challenging for readers to follow. Numbering the equations could enhance readability.
It has been done as requested. - Is it possible to ensure uniformity in the font size in Figure 1, specifically addressing the discrepancy at line 207? Additionally, could the authors provide an explanation for the representations 'a' and 'b' in Figure 1, and clarify the meaning of 'P≤0.05'?
It has been done as requested. - Could the authors consider introducing abbreviations at their first instance in the manuscript, for instance, in Section 3.2, to ensure consistency and aid reader comprehension throughout the document?
It has been done as requested. - In Table 1, it would be helpful for the authors to explicitly define each abbreviation, akin to the approach taken in other tables, such as 'n.d' at the end of the table.
Thank you for your comment. We corrected at the table. - Clarification on the meaning of 'p' and the p-values in Tables 2 and 3, respectively would be beneficial to avoid reader confusion and maintain consistency.
It has been done as requested. - Regarding the caption of Figure 5 at line 343, could 'means' be adjusted to 'mean' for accuracy?
It has been done as requested. - The manuscript's language must be reviewed to ensure it caters to a broader readership.
Thank you for your comment. It will be reviewed by the language editing by MDPI
Round 2
Reviewer 1 Report
Comments and Suggestions for Authors
The authors made the suggested changes. Therefore, it can be considered for publication.